# 25-Hydroxyvitamin D Measurement in Human Hair: Results from a Proof-of-Concept study

**DOI:** 10.3390/nu11020423

**Published:** 2019-02-18

**Authors:** Lina Zgaga, Eamon Laird, Martin Healy

**Affiliations:** 1School of Medicine, Trinity College Dublin, The University of Dublin, Dublin 2, Ireland; lairdea@tcd.ie; 2Department of Biochemistry, St James’s Hospital, Dublin 8, Ireland; mhealy@stjames.ie

**Keywords:** vitamin D, 25-hydroxyvitamin D, hair, vitamin D status assessment

## Abstract

Vitamin D deficiency has been implicated in numerous human diseases leading to an increased interest in assessing vitamin D status. Consequentially, the number of requests for vitamin D measurement keeps dramatically increasing year-on-year. Currently, the recognised best marker of vitamin D status is the concentration of the 25-hydroxyvitamin D (25(OH)D_3_) in the blood circulation. While providing an accurate estimate of vitamin D status at the point in time of sampling, it cannot account for the high variability of 25(OH)D_3_ concentration. In this proof of concept study we set out to provide evidence that 25(OH)D_3_ can be extracted from hair samples in a similar fashion to steroid hormones. Two of the authors (L.Z. and M.H.) provided hair samples harvested from the crown area of the scalp and the third author (E.L.) provided beard samples. These samples, cut into 1 cm lengths, were weighed, washed and dried. 25(OH)D was extracted using a previously published steroid hormones extraction procedure. Blood samples were taken from the subjects at the same time all tissue samples were analysed using liquid-chromatography mass spectrometry. Hair samples showed presence of quantifiable 25(OH)D_3_ with concentrations ranging from 11.9–911 pg/mg. The beard sample had a concentration of 231 pg/mg. Serum levels of 25(OH)D_3_ ranged from 72–78 nmol/L. The results presented here confirm the feasibility of measuring 25(OH)D_3_ in hair samples. The findings warrant further validation and development and have the potential to yield valuable information relating to temporal trends in vitamin D physiology.

## 1. Introduction

Vitamin D deficiency has been linked with a wide range of diseases, including bone health, cancer, susceptibility to infections, inflammation and autoimmune, cardiovascular, and metabolic diseases [1,2]. The main sources of vitamin D for humans are dermal synthesis in skin following exposure to sunlight [3,4] and vitamin D supplements [5,6]; unless fortified, dietary sources are virtually negligible for the majority of people [7,8]. Because of our indoor oriented lifestyle, clothing, use of skin products (that block UVB light), pollution, etc. the opportunity for skin synthesis is reduced and vitamin D deficiency is a world-wide epidemic with over 1 billion people estimated to be deficient [9].

Once vitamin D is ingested or synthesised in the skin, it is hydroxylated in the liver to 25-hydroxyvitamin D (25(OH)D). This form of vitamin D is most abundant in the human body, and acts as a storage form of vitamin D. Currently, the internationally recognised way of assessing vitamin D status is to measure the concentration of circulating 25(OH)D in blood samples. There are several methods available for achieving this. Liquid chromatography-mass spectrometry (LC-MS/MS) is regarded as the gold standard and has the advantages of sensitivity, specificity and the ability to resolve 25(OH)D_3_, the physiological form of the vitamin from its plant-based isomer 25(OH)D_2_ [10,11,12]. Immunochemistry assays are a commonly used alternative for 25(OH)D estimation but have significant drawbacks, for example the inability to fully detect and quantitate the 25(OH)D_2_ isomer and cross-reactivity with other metabolites of vitamin D such as 24,25(OH)2D. 

Analysis of blood samples for vitamin D is not without disadvantages. It is invasive, requires expertise, training, and appropriate equipment and hygienic conditions, so obtaining samples frequently—or in some cases at all—is not always workable. In addition, the resulting concentration represents a single point in time of vitamin D standing and gives limited information on an individual’s long-term status. 

All forms of vitamin D belong to the secosteroid family in which a bond (C9–C10) in ring B of the steroid structure is broken. Secosteroids share structural similarities with steroid hormones such as cortisol and testosterone. The biologically active form of vitamin D, 1,25-dihydroxyvitamin D, is considered a steroid hormone because of its actions in binding and activating the nuclear vitamin D receptor. Hair testing has been available for several years in forensic testing and for the analysis of steroids [13,14,15,16]. It has several advantages over traditional matrices in that it is non-invasive, sample is easily stored and transported, and is very stable over long periods of time even without sample preparation or refrigeration. Moreover, hair samples can provide a longitudinal view of chosen biomarker. This approach of analysing hair samples has not been applied to vitamin D. 

We carried out a proof-of-concept study to provide preliminary data on the feasibility of extracting and measuring vitamin D in hair samples. To the best of our knowledge vitamin D analysis in hair samples has not been previously reported in the literature.

## 2. Materials and Methods

### 2.1. Subjects

The subjects included the three study authors. Initially, L.Z. provided a hair sample (brown head hair) in March 2018 (Study 1). Following a successful detection of the vitamin, the experiment was replicated in July 2018 (Study 2) when three subjects provided hair and blood samples. All three subjects provided informed consent to participate in the study; ethical approval was not required because investigators themselves provided all of samples. 

### 2.2. Hair Collection, Washing and Sample Preparation

The hair collection procedure followed the consensus recommendations provided by the Society of Hair Testing [17]. L.Z. and M.H. provided hair samples, approximately 14 cm and 4 cm long, respectively. The hair (dry) was cut as close as possible from posterior vertex region of the scalp. Samples were cut into 1.25 cm segments analysed separately; L.Z. hair was cut into 11 segments and M.H. hair sample into 4 segments. E.L. provided the beard sample (2 bundles of 4 cm length). All samples were given an identification number, weighed, and stored in a dry and dark environment at room temperature until ready for transport to the laboratory for analysis. 

This proof of concept investigation was instigated to determine if vitamin D (in the form of 25-hydroxyvitamin D_3_) was present in hair samples in measurable amounts. We reviewed the literature and found no reports on extraction of vitamin D from hair samples. Because of the secosteroid structure of vitamin D we employed a previously developed procedure for steroid hormone extraction from hair [18]. 

The first step in sample preparation involved washing, which decontaminates the outer surface of the hair by removing non-blood-borne steroids. Washing also prevents analytical interferences from external sources, such as hair care products, sweat, sebum, etc. Samples were individually washed by vortexing in 2.0 mL isopropanol for two minutes at room temperature to remove contaminants. Samples were subsequently dried using lint-free tissue and cut into smaller strips. These were transferred to glass test tubes and 25(OH)D was extracted by adding 1.4 mL mass spectrometry grade methanol and mixing on a roller mixer for 18 h. Following the extraction period samples were spun at 1500 *g* for 15 min. The methanol extracts were transferred to new tubes and evaporated under nitrogen stream at 50 °C until dry. Samples were re-suspended in 120 µL methanol for assay by LC-MS/MS. Blank samples containing no hair and brought through the extraction process were included in the analyses. In the isopropanol cleaning step aliquots of the wash solution were taken for analysis to determine if 25(OH)D leaching occurred from hair samples during this step. 

### 2.3. Blood Sample Collection

A blood sample (7 mL, non-fasting) was collected by venepuncture into an evacuated tube by a trained phlebotomist. Samples were kept chilled and centrifuged (1500 *g* for 15 min) within 3 h of collection and serum aliquots were labelled and stored at −80 °C until required for analysis. 

### 2.4. 25(OH)D Assessment 

Both serum and re-suspended hair extracts were assayed using a validated mass spectrometry method developed by Chromsystems Instruments and Chemicals GmbH, Munich (Mass Chrom-25-OH-Vitamin D_3_/D_2_) using tandem liquid chromatography mass spectrometry (API 4000; SCIEX, Framingham, MA, USA) and analysed in the Biochemistry Department of St. James’s Hospital, Dublin, Ireland. The laboratory is ISO 15189 accredited. Visualisation of vitamin D chromatography and calculation of concentrations is performed by propriety software (Analyst 1.6.1: Sciex, Framingham, MA, USA). Assay quality was monitored using in-house and third-party quality controls. Accuracy was determined using the National Institute of Standards and Technology vitamin D standard reference material 972. The laboratory also participates in the Vitamin D External Quality Assessment Scheme (DEQAS). The respective inter- and intra-assay coefficients of variation were 5.7% and 4.5%. 

### 2.5. Approximation of 25(OH)D Concentration in Hair

The following equation [19] was used to approximate the concentration (pg/mg) of 25OHD_3_ in hair samples:$$c\left(\frac{\mathrm{pg}}{\mathrm{mg}}\right)\left(\mathrm{hair}\right)=\frac{c\left(\mathrm{calculated}\right)\left[\frac{\mathrm{ng}}{\mathrm{mL}}\right]\times V1\left(\mathrm{extraction}\right)\left[\mathrm{mL}\right]}{\frac{V2\left(\mathrm{evaporation}\right)\left[\mathrm{mL}\right]}{V3\left(\mathrm{reconstitution}\right)\left[\mathrm{mL}\right]}\times m\left(\mathrm{sample}\right)\left[\mathrm{mg}\right]}\times 1000$$
where c (calculated) is the concentration of 25(OH)D_3_ obtained by LC-MS/MS in nmol/L ÷ 2.5 (to convert to ng/mL); V1 (extraction volume = 1.4 mL); V2 (evaporation volume = 1.4 mL); V3 (reconstitution volume = 120 µL (0.120 mL)).

## 3. Results

The characteristics of subjects and samples are presented in Table 1 and Table 2. One female and two males were included; all participants were normal weight and reported taking vitamin D supplements (1000 IU D_3_ daily). All subjects were vitamin D sufficient (circulating 25OHD_3_ > 50 nmol/L). 

In Study 1, one hair sample (bunched sample of 75 mg) was analysed by LC-MS/MS and 25(OH)D_3_ was detected. The chromatogram for the initial exploratory study demonstrated a clear signal for the presence of vitamin D in the hair sample is shown in Figure 1 (the absence of a 25(OH)D_3_ peak in the blank sample was noted). 

In Study 2, the experiment was repeated in 15 hair and one beard sample and in blood samples. Weight of hair samples varied between 20 and 44 mg (median 28 mg; mean 29.38 ± 6.88 (SD) mg). LC-MS/MS method detected 25(OH)D_3_ in hair samples from all subjects. Virtually no 25(OH)D_2_ was observed in any of the samples. A large variation was observed in measurements. 25(OH)D_3_ concentrations were low in all L.Z. samples, with 25(OH)D_3_ undetectable in some hair segments. Average approximated 25(OH)D_3_ concentration in L.Z. samples was 19 pg/mg (highest 30.9 pg/mg). In contrast, in M.H. samples average approximated concentration was 421 pg/mg (lowest 26.5 and highest 911 pg/mg, Figure 2. In the E.L. beard sample, concentration was approximated to be 231 pg/mg (Figure 3). 

## 4. Discussion

As far as the authors are aware of this is the first published study to demonstrate that 25(OH)D_3_ can be found in human hair and beard hair samples. Serum concentrations of 25(OH)D_3_ in the study subjects averaged 75 nmol/L. Hair concentrations of the vitamin had a median concentration of 28.7 pg/mg in all samples tested although there was a significant variance in levels between L.Z. and M.H.. There is limited evidence for a direct relationship between serum and hair concentrations of steroid hormones and the mechanisms by which metabolites enter hair are not fully understood. Both steroid hormones and 25(OH)D_3_ differ in the dynamics of their circulation on a daily basis and traditional blood serum analysis is a snapshot that yields little temporal information of the status of metabolic products. In contrast segmental analysis of hair, which grows at approximately 1 cm/month, provides an historical blueprint of the systemic status of hormone circulation over a period of months [20] and could reflect 25(OH)D_3_ status over an extended period depending on the length of hair sampled. Large variation in status, primarily responding to varying solar intensity (Figure 4), but also supplement intake could be monitored on a longitudinal basis [3,4]. 

Previous work in this area has involved measuring a range of minerals in hair and then using computational modelling to estimate the vitamin D status [21,22]. Although this can give an estimate, it does not measure the actual concentrations of vitamin D in the hair itself. Within the current study, we have used a mathematical formula that takes the measured concentrations obtained by LC-MS/MS and approximates the concentration in the hair [18]. While serum concentrations were in the sufficient range and quite similar in all study subjects, we have observed a large variation between subjects and some variation within subjects of hair 25(OH)D_3_ levels. Further research is needed to establish whether 25(OH)D deposition in hair is reflective of 25(OH)D in the blood at the time of deposition, and also to investigate what other factors affect the 25(OH)D in hair: concentration might be affected by other (steroid) hormones (and hence it might differ between males and females); colour or other hair characteristics, behaviours such as time spent in the sun (photodegradation) [23] and other personal characteristics and behaviours may all affect the deposition and/or degradation of 25(OH)D in hair. Factors that might affect the extraction process also need to be investigated. The lack of correlation between the blood concentrations and those measured in the hair is not unexpected. While blood concentration is a point-measurement of vitamin D status, hair provides a period-measurement with lag and hence may not correlate significantly with blood levels (Figure 4). Hair analysis, therefore, of 25(OH)D_3_ is not seen as a replacement of blood measurements but an adjunct with the ability of providing more information on long term vitamin D status—‘the tree-ring effect’. This would be particularly useful for long-term epidemiological research, longitudinal studies and for long duration vitamin D randomized controlled trials where hair could be used to detect compliance with supplementation. 

Additional advantages that relate to the proposed hair sampling procedure are: (*i*) it would be painless as no venepuncture is necessary—this is particularly important for some subgroups, most notably children, other vulnerable individuals, and those in need of repeated testing or in a setting where phlebotomy is not readily accessible. (*ii*) it is simpler and does not require the involvement of a professional (such as a nurse) or an array of consumables (tubes, needles, alcohol wipes etc.). (*iii*) the sample handling is risk-free (e.g., blood born infections). (*iv*) samples can be safely despatched from anywhere by post—this is in contrast to specialist shipment required for blood preparations. (*v*) hair samples for analysis are extremely stable.

Other applications of the method could also involve analysis of historical samples from archaeological sites. Hair (along with teeth) are some of the longest lasting surviving biological materials after death [24] and thus it could be possible to for the first time assess the vitamin D status of historical populations—Elizabethans, Viking, Celtic, Roman, ancient Chinese, Egyptian, bronze age etc. and even possibly older humans if any surviving hair can be found in the permafrost. New insights in population genetics might be gained, as the evolution of lighter skin colour is hypothesised to be linked with vitamin D deficiency [25]. Similarly, hair samples could also be used to assess longer-term vitamin D status in animals with applications in farming, for example improving fertility [26]. The vitamin D status of ancient species could be measured given the well preserved and copious amounts of, for example, mammoth or ancient ice age animal hair that is often unearthed from the warming permafrost and in museum specimens.

To the best of our knowledge, this is the first study to measure and successfully demonstrate the presence of 25(OH)D in hair samples. We replicated our first pilot study with additional samples, we measured different hair types, colours and different gender and ages of participants. LC-MS/MS was used for 25(OH)D detection; this method has very high sensitivity and specificity even at small concentrations and is recognised as the gold standard of vitamin D assessment [12]. Our study also has some limitations. In particular the small sample size, a limited number of measurements, and lack of variability of serum vitamin D concentration between subjects. Furthermore, we also had no data on past vitamin D status and sun exposure. The LC-MS method employed in this study does not resolve inactive epimers of 25(OH)D_3_ although their circulating concentrations are very low and add negligibly to overall vitamin D status in non-neonatal samples [27].

Larger studies encompassing assay optimisation and validation, greater variability in 25(OH)D_3_ status, different age and ethnic groups, multiple blood measurements and information on a range of other relevant factors are needed to further our understanding of the relationship between circulating 25(OH)D levels and concentration in hair. Concentration in hair segments for which an exact growth time is known has the potential to provide useful information on variations in 25(OH)D_3_ concentration over that time period, but further research is needed. 

## 5. Conclusions

In conclusion, we confirmed the proof of concept that 25(OH)D_3_ can be detected in human hair and beard samples. Further research is needed to clarify the relationship between circulating concentration and 25(OH)D_3_ found in hair in different population groups, and to estimate levels in different age groups, genders, ethnicities, animals and historical samples. Given the high proportion of vitamin D deficiency across the world and the ever-increasing demand for 25(OH)D_3_ blood testing, a non-invasive method that captures vitamin D status over a prolonged time period has the potential to add significantly to vitamin D epidemiological studies. In addition, where phlebotomy is not readily available, a hair sample would make a useful substitute to determine vitamin D status. 

## Figures and Tables

**Figure 1 nutrients-11-00423-f001:**
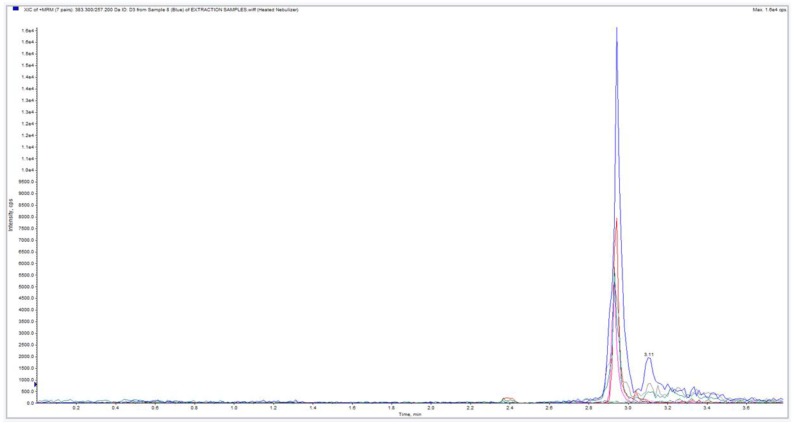
Chromatogram of hair sample in Study 1. Large blue peak represents hair extracted 25(OH)D_3_. Red peak is the assay 25(OH)D_3_ internal standard.

**Figure 2 nutrients-11-00423-f002:**
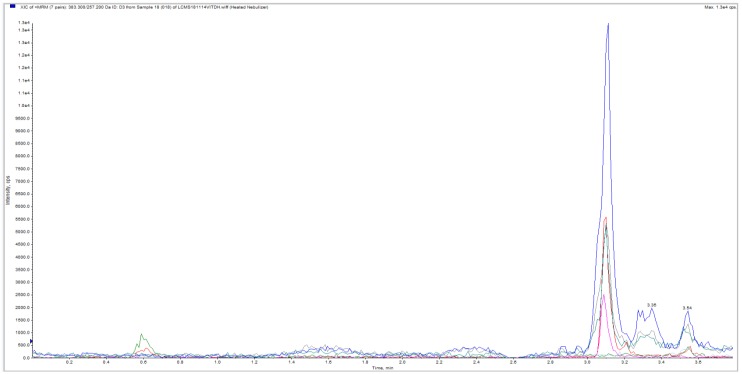
Chromatogram of hair sample from M.H. in Study 2. Large blue peak represents hair extracted 25(OH)D_3_. Red peak is the assay 25(OH)D_3_ internal standard. M.H., Author initial.

**Figure 3 nutrients-11-00423-f003:**
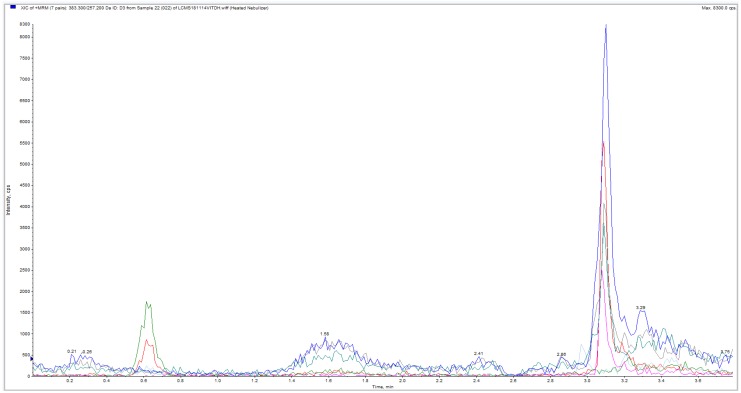
Chromatogram of beard hair sample from E.L. in Study 2. Large blue peak represents beard hair extracted 25(OH)D_3_. Red peak is the assay 25(OH)D_3_ internal standard. E.L., Author initial.

**Figure 4 nutrients-11-00423-f004:**
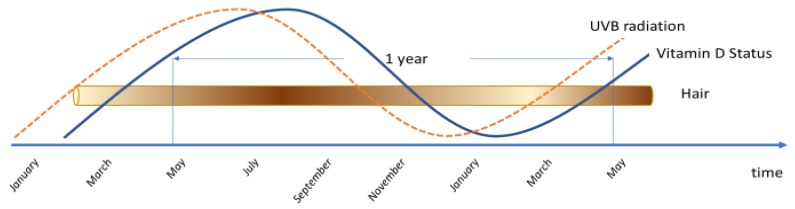
Large seasonal changes in ultraviolet-B (UVB) radiation (- - -), vitamin D status (full line, lags behind UVB) and theoretical deposits of vitamin D in the hair (varied shading intensities) are shown.

**Table 1 nutrients-11-00423-t001:** Subject characteristics.

Subject	Gender	Age	Hair Color	BMI	Circulating 25OHD (nmol/L)
1	Female	37	Brown	22.5	72
2	Male	64	Grey	21.5	76
3	Male	33	Ginger	24.9	78

**Table 2 nutrients-11-00423-t002:** Samples analysed in Study 2.

Sample ID	Subject	Sample	Segment	Weight (mg)	LC-MS/MS	Approximated Sample Conc. (pg/mg)
1	1 (L.Z.)	hair	1 (root)	27	*ND*	*ND*
2			2	27	*ND*	*ND*
3			3	29	+	20.3
4			4	25	*ND*	*ND*
5			5	23	*ND*	*ND*
6			6	20	*ND*	*ND*
7			7	24	+	16.2
8			8	26	+	13.6
9			9	30	+	30.9
10			10	40	*ND*	*ND*
11			11 (end)	44	+	11.9
12	2 (M.H.)	hair	1 (root)	35	+	433
13			2	35	+	26.5
14			3	30	+	315
15			4 (end)	20	+	911
16	3 (E.L.)	beard	1	35	+	231
17	Blank				-	0

Abbreviations: LC-MS/MS, liquid chromatography tandem mass spectrometry. +, positive detection of 25(OH)D. ND, No 25(OH)D peak detected. L.Z., M.H., E.L., Author initials.

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
