# Peer review of "25-Hydroxyvitamin D Measurement in Human Hair: Results from a Proof-of-Concept study"

_nutrients, 2019, doi:10.3390/nu11020423_

Round 1

Reviewer 1 Report

The authors describe a new method for detecting 25 (OH) D concentration in human hair. I do not know that this has been done before. If the method can be developed and replicated, the method can probably be interesting in order to obtain a complementary picture of the vitamin D status.

There are several pure errors of fact and ambiguity in the article that need to be corrected. However, I believe that the article may be of interest to many who are interested in vitamin D research.

In abstract:

Line 22. You write

 “We detected 25(OH)D in hair and beard samples 22 from all subjects and the concentration was approximated”

I believe this is not correct. In the method part you write it was hair subjects only from LZ and MH and beard samples from EL?

It would be interesting for the reader to also obtain information in the abstract about the concentration of 25 (OH) D in blood and hair, respectively. There are no results at all in the abstract.

In Introduction

Line 45. You writeEnzyme-linked immunosorbent assays (ELISA) are the most common alternative”

This is not correct. The most common methods nowadays are not ELISA methods. It is immunochemistry methods.

Line 52. Observe that it is only the vitamin D metabolite 1,25 (OH) 2D which is considered a hormone and which affects the vitamin D receptor.

Material and methods

Line 82 you write vitamin D, should be 25(OH)D

Line 88 you write vitamin D, should be 25(OH)D

Line 119 you write signal for presence of vitamin D. Should be signal for presence of 25(OH)D3

Line 124 you write 25(OH)D. Should  be 25(OH)D3

Line 142 you write 25(OD)D3. Should be 25(OH)D3

Line 143 similar wrong spelling you write 25(OD)3. Should be 25(OH)D3

Line 148. You write “arguably the first big vitamin D methodological leap in over a decade” I would suggest you to remove this. There are many major methodological discoveries in recent years. For example, free vitamin D etc.

Line 151. You write about cortisol. There are large differences between the concentration in cortisol blood compared to the concentration of 25 (OH) D. Cortisol has a pronounced and large daily variation. 25 (OH) D, on the other hand, has a half-life of over one month.

Line 206 you write 25(OH)D. Should be 25(OH)D3

Line 217. You write “a very accurate estimate” It is possible that the method can be developed and become very accurate but you do not show it in this study

Line 211. You write about the inactive epimer of 25(OH)D3. It depends on your LC-MS/MS method if this epimer can be detected with your method. Check this.

Line 153 and line 154 and more. You constantly mix units in molar units, mass units (grams), SI units and conventional units. It would be much better if you could be a little more consistent.

Author Response

Reviewer 1

The authors describe a new method for detecting 25 (OH) D concentration in human hair. I do not know that this has been done before. If the method can be developed and replicated, the method can probably be interesting in order to obtain a complementary picture of the vitamin D status. There are several pure errors of fact and ambiguity in the article that need to be corrected. However, I believe that the article may be of interest to many who are interested in vitamin D research.

Author response: We thank reviewer 1 for their feedback and useful comments on our manuscript. We have addressed the concerns and our responses to each point made are listed below.

1) Line 22. You write “We detected 25(OH)D in hair and beard samples 22 from all subjects and the concentration was approximated”. I believe this is not correct. In the method part you write it was hair subjects only from LZ and MH and beard samples from EL?

Response: Hair samples were obtained from LZ and MH and a beard sample from EL. This sentence has been re-worded to make this clearer in lines 17-18:

“Two of the authors (LZ and MH) provided hair samples harvested from the crown area of the scalp and the third author (EL) provided beard samples.”

2) It would be interesting for the reader to also obtain information in the abstract about the concentration of 25 (OH) D in blood and hair, respectively. There are no results at all in the abstract.

Response: We agree with the reviewer. Details regarding the concentrations found have now been added to the abstract in lines 22-24:

“Hair samples showed presence of quantifiable 25(OH)D3 with concentrations ranging from 11.9-911 pg/mg. The beard sample had a concentration of 231 pg/mg. Serum levels of 25(OH)D3 ranged from 72-78 nmol/l.”

3) Line 45. You write “Enzyme-linked immunosorbent assays (ELISA) are the most common alternative”. This is not correct. The most common methods nowadays are not ELISA methods. It is immunochemistry methods.

Response: We have changed the wording (lines 46-48) to:

“Immunochemistry assays are a commonly used alternative for 25(OH)D estimation but have significant drawbacks, for example the inability to fully detect and quantitate the 25(OH)D2 isomer and cross-reactivity with other metabolites of vitamin D such as 24,25(OH)2D.”

4) Line 52. Observe that it is only the vitamin D metabolite 1,25 (OH) 2D which is considered a hormone and which affects the vitamin D receptor.

Response: We have modified lines 54-58 to address this comment:

“‘All forms of vitamin D belong to the secosteroid family in which a bond (C9–C10) in ring B of the steroid structure is broken. Secosteroids share structural similarities with steroid hormones such as cortisol and testosterone. The biologically active form of vitamin D, 1,25-dihydroxyvitamin D, is considered a steroid hormone because of its actions in binding and activating the nuclear vitamin D receptor.”

5) Line 82 you write vitamin D, should be 25(OH)D

Response: We have changed this to 25(OH)D.

6) Line 88 you write vitamin D, should be 25(OH)D

Response: We have changed this to 25(OH)D.

7) Line 119 you write signal for presence of vitamin D. Should be signal for presence of 25(OH)D3

Response: We have changed this to 25(OH)D3.

8) Line 124 you write 25(OH)D. Should  be 25(OH)D3

Response: We have changed this to 25(OH)D3.

9) Line 142 you write 25(OD)D3. Should be 25(OH)D3

Response: We have changed this to 25(OH)D3.

10) Line 143 similar wrong spelling you write 25(OD)3. Should be 25(OH)D3

Response: We have changed this to 25(OH)D3.

11) Line 148. You write “arguably the first big vitamin D methodological leap in over a decade” I would suggest you to remove this. There are many major methodological discoveries in recent years. For example, free vitamin D etc.

Response: We have removed this statement.

12) Line 151. You write about cortisol. There are large differences between the concentration in cortisol blood compared to the concentration of 25 (OH) D. Cortisol has a pronounced and large daily variation. 25 (OH) D, on the other hand, has a half-life of over one month.

Response: While we agree with the reviewer that there is a difference between cortisol and vitamin D, it was used as an analogy to demonstrate the principle of the idea that as a group, there is limited correlation between steroid hormone blood concentration and steroid hair concentration. In lines 161-165 we have added: “There is limited evidence for a direct relationship between serum and hair concentrations of steroid hormones and the mechanisms by which metabolites enter hair are not fully understood. Both steroid hormones and 25(OH)D3 differ in the dynamics of their circulation on a daily basis and traditional blood serum analysis is a snapshot that yields little temporal information of the status of metabolic products.”

13) Line 206 you write 25(OH)D. Should be 25(OH)D3.

Response: We have changed this to 25(OH)D3.

14) Line 217. You write “a very accurate estimate”. It is possible that the method can be developed and become very accurate but you do not show it in this study.

Response: Yes, we agree with the reviewer and have clarified this statement in lines 231-233 “Concentration in hair segments for which an exact growth time is known has the potential to provide useful information on variations in 25(OH)D3 concentration over that time period, but further research is needed.”

15) Line 211. You write about the inactive epimer of 25(OH)D3. It depends on your LC-MS/MS method if this epimer can be detected with your method. Check this.

Response: We have clarified this statement in lines 225-227: “The LC-MS method employed in this study does not resolve inactive epimers of 25(OH)D3 although their circulating concentrations are very low and add negligibly to overall vitamin D status in non-neonatal samples [27].”.

16) Line 153 and line 154 and more. You constantly mix units in molar units, mass units (grams), SI units and conventional units. It would be much better if you could be a little more consistent.

We agree with the reviewer. However, there is a lack of standardisation in expressing units in the vitamin D literature in general and this can lead to confusion – for example, vitamin D intake is reported in IU or micrograms; circulating levels in ng/mL or nmol/L. We have commented on the variation of hair 25(OH)D3 within our sample and mentioned the lack of evidence on the relationship between circulating steroids and hair concentrations. This needs to be addressed in future studies in relation to 25(OH)D3.

Reviewer 2 Report

Overall well written and clear article.  Subject matter is interesting and adds valuable new data to the field of Vit. D research. 

My one major concern is in Section 2.2 Sample preparation.  I would like additional details regarding Vit. D extraction from the hair.  Methanol was used to extract the Vit. D.  Is this an established method for high yield extraction of Vit. D (or other steroids) from hair?  If so, please add a reference.  If this is an extraction method developed by the authors, please give more detail explaining your confidence in this extraction technique.  Could any other solvents have been used?  Are there any other methods in the literature for Vit. D extraction?  Were any other extraction methods tried by the authors?  If so, describe why methanol for 18 hours is the best method.

Author Response

Reviewer 2

Overall well written and clear article.  Subject matter is interesting and adds valuable new data to the field of Vit. D research. 

Response: We thank the reviewer for reading our manuscript and for providing valuable feedback.

1). Comment: My one major concern is in Section 2.2 Sample preparation.  I would like additional details regarding Vit. D extraction from the hair.  Methanol was used to extract the Vit. D.  Is this an established method for high yield extraction of Vit. D (or other steroids) from hair?  If so, please add a reference.  If this is an extraction method developed by the authors, please give more detail explaining your confidence in this extraction technique.  Could any other solvents have been used?  Are there any other methods in the literature for Vit. D extraction?  Were any other extraction methods tried by the authors?  If so, describe why methanol for 18 hours is the best method.

Response: As a proof-of-concept investigation, we wanted initially to see if we could establish that vitamin D (in the form of 25-hydroxyvitamin D) was indeed present in hair samples in measureable amounts. We carefully reviewed the literature and found no reports on extraction of vitamin D from hair samples. There are, however, numerous studies on analysis of steroid hormones in hair. These include cortisol and testosterone. Because vitamin D is a secosteroid and related to the steroid hormone family in structure, we decided to initially use a published method for steroid hormone extraction from hair. However, there is little standardisation in the extraction approach for steroids with several different solvents used and varying lengths of incubation. What consensus there was suggested that isopropanol was suitable for the washing step and methanol was the most useful for extraction with an 18-hour incubation period, so we chose this approach for our first experiment.  Our research question for the project was to determine whether vitamin D can be detected in hair, and we have successfully shown this was the case when using isopropanol for washing and methanol for extraction – therefore, we have so far not tried other methods. Having said this, we absolutely agree with the reviewer that this needs to be investigated further in future validation work. Given the positive findings presented here, we are planning a more thorough examination of washing and extraction procedures.

Round 2

Reviewer 1 Report

I think the paper has improved significantly now. It is an interesting article.